# High prevalence and extended deletions in *Plasmodium falciparum hrp2/3* genomic loci in Ethiopia

**Lemu Golassa** [1]*, **Alebachew Messele**[1], **Alfred Amambua-Ngwa**[2], **Gote Swedberg**[3]

**1** Aklilu Lemma Institute of Pathobiology, Addis Ababa University, Addis Ababa, Ethiopia, **2** MRC Unit The Gambia at the London School of Hygiene and Tropical Medicine, Banjul, The Gambia, **3** Department of Medical Biochemistry and Microbiology, Uppsala University, Uppsala, Sweden

\* lgolassa@gmail.com

**Data Availability Statement:** All relevant data are within the paper and its Supporting Information files.

**Funding:** This work was supported through the DELTAS Africa Initiative [DELGEME grant 107740/

## Abstract

Deletions in *Plasmodium falciparum* histidine rich protein 2(*pfhrp*2) gene threaten the usefulness of the most widely used HRP2-based malaria rapid diagnostic tests (mRDTs) that cross react with its structural homologue, *Pf*HRP3. Parasites with deleted *pfhrp*2/3 genes remain undetected and untreated due to 'false-negative' RDT results. As Ethiopia recently launched malaria elimination by 2030 in certain selected areas, the availability of RDTs and the scale of their use have rapidly increased in recent years. Thus, it is important to explore the presence and prevalence of deletion in the target genes, *pfhrp*2 and *pfhrp*3. From a total of 189 febrile patients visited Adama Malaria Diagnostic centre, sixty-four microscopically- and polymerase chain reaction (PCR)-confirmed *P. falciparum* clinical isolates were used to determine the frequency of *pfhrp*2/3 gene deletions. Established PCR assays were applied to DNA extracted from blood spotted onto filter papers to amplify across *pfhrp*2/3 exons and flanking regions. However, analysis of deletions in *pfhrp*2, *pfhrp*3 and flanking genomic regions was successful for 50 of the samples. The *pfhrp*2 gene deletion was fixed in the population with all 50(100%) isolates presenting a deletion variant. This deletion extended downstream towards the Pf3D7 0831900 (MAL7PI.230) gene in 11/50 (22%) cases. In contrast, only 2/50 (4%) of samples had deletions for the Pf3D7 0831700 (MALPI.228) gene, upstream of *pfhrp*2. Similarly, the *pfhrp*3 gene was deleted in all isolates (100%), while 40% of the isolates had an extension of the deletion to the downstream flanking region that codes for Pf3D7 13272400 (MAL13PI.485).The *pfhrp*3 deletion also extended upstream to Pf3D7 081372100 (MAL13PI.475) region in 49/50 (95%) of the isolates, exhibiting complete absence of the locus. Although all samples showed deletions of *pfhrp*2 exon regions, amplification of an intron region was successful in five cases. Two different repeat motifs in the intron regions were observed in the samples tested. *Pfhrp*2/3 gene deletions are fixed in Ethiopia and this will likely reduce the effectiveness of *Pf*HRP2-based mRDTs. It will be important to determine the sensitivity *Pf*HRP 2/3-based RDTs in these populations and conduct a countrywide survey to determine the extent of these deletions and its effect on routine RDT-based malaria diagnosis.

Z/15/Z]. The DELTAS Africa Initiative is an independent funding scheme of the African Academy of Sciences (AAS)'s Alliance for Accelerating Excellence in Science in Africa (AESA) and supported by the New Partnership for Africa's Development Planning and Coordinating Agency (NEPAD Agency) with funding from the Wellcome Trust [DELGEME grant 107740/Z/15/Z] and the UK government. The funders had no role in study design, data collection and analysis, decision to publish, or preparation of the manuscript.

**Competing interests:** This work was supported in part by the Wellcome Trust. This does not alter our adherence to PLOS ONE policies on data or materials sharing.

**Abbreviations:** DNA, Deoxyribonucleic acid; GTS, Global Technical Strategy; Hrp2/3, Histidine rich protein 2/3; NMCP, National malaria control program; *Pf*HRP2/3, *Plasmodium falciparum* histidine rich protein 2; nPCR, Nested polymerase chain reaction; PCR, Polymerase chain reaction; pLDH, Plasmodium lactate dehydrogenase; mRDT, Malaria rapid diagnostic test; RDTs, Rapid diagnostic tests; RNA, Ribonucleic acid; rRNA, Ribosomal RNA; WHO, World Health Organization.

## Introduction

In Ethiopia, *Plasmodium falciparum* and *Plasmodium vivax* are co-transmitted and respectively accounted for 60% and 40% of all malaria cases [1,2]. Nearly 68% of the landmass of Ethiopia is favourable for malaria transmission [1] and endemicity is heterogeneous with varied epidemiological presentation in different geographic settings [3]. Like in many parts of Africa, the incidence of malaria has substantially declined with a reported 40% reduction between 2000 and 2015 [4,5]. Ethiopia is on track to achieve the 2020 milestone to reduce the incidence of malaria by 40%. This also aligns with the World Health Organization (WHO) Global Technical Strategy (GTS) to intensify existing malaria interventions towards elimination by 2030 [5].

Rapid diagnostic tests (RDTs) were also introduced to improve early diagnosis of malaria in remote areas where microscopic examination of blood smears remains impractical. A majority of commercially available mRDTs are designed to detect malaria specific antigens such as lactate dehydrogenase or aldolase for pan-malaria diagnosis and *Pf*HRP2 for *P. falciparum* specific diagnosis [6]. RDTs have become extremely essential for implementing early diagnosis and prompt effective treatment of malaria and for the continuous reduction of its burden.

In Ethiopia, mRDTs were introduced as one of the diagnostic methods following the revision of malaria diagnosis and treatment guideline in 2004 in the country. Depending on the antigen they target, different types of RDTs exist. Those that target histidine-rich protein-2 (HRP-2) only detect *P. falciparum*, while those that target the parasite enzyme lactate dehydrogenase (LDH) and aldolase can detect non-falciparum from mixed infection [7,8]. PfHRP2/3-based RDTs have been widely used for detection of *P. falciparum* at health posts/community levels in Ethiopia since 2005 [9].

*Pf*HRP2 is a non-essential protein encoded by *pfhrp2* gene located on chromosome 8 of *P. falciparum*. Its structural homologue, *Pf*HRP3, is coded by a locus on chromosome 13 [10]. *Pfhrp3* antigen epitopes are recognised by some *Pf*HRP2-based RDTs [11] and may influence the diagnostic performance of these mRDTs. Hence, *Pf*HRP3 contributes to reactivity of *Pf*HRP2-based RDTs. Although *Pf*HRP2-based RDTs have been widely used, its performance is complicated by the natural deletion of *pfhrp2* gene in parasite populations in some geographical regions. Variation in the performance of RDTs has been observed, probably driven by polymorphisms in gene loci targeted, such as the recently described deletions in the *Pfhrp2/3* loci. The prevalence and dynamics of these *pfhrp2/3* deleted *P. falciparum* strains and their impact on diagnosis has not been extensively investigated in Ethiopia. Following the first report detailing the deletion of the *pfhrp2* gene in *P. falciparum* isolates from Peru, several studies have shown the global spread of malaria parasites lacking *pfhrp2* gene and the flanking chromosomal regions [12]. This generated anxiety on possible reduced sensitivity of *Pf*HRP2-based RDTs. In Eritrea, *P. falciparum* lacking *pfhrp2* now constitute a major threat to malaria control [13,14] as they are not detected by *Pf*HRP2-based RDTs and remain untreated. It has been suggested that these strains with deletions at *pfhrp2/3* genes have a fitness advantage and pose a challenge to progress made in malaria control and elimination [15] as these parasites will escape detection by *Pf*HRP2-based RDTs and may be selected to expand due to routine use of RDTs leading to increasing frequencies of parasite population with *pfhrp2/3* deleted genes in the communities. Though *P. falciparum* strains without these loci continue to thrive, the role of *Pf*HRP2/3 loci in parasite virulence and fitness is not clear as these are expressed in all stages of development of parasite, probably contributing to survival advantage [16–18].

A substantial proportion of parasite isolates with both *pfhrp2* and *pfhrp3* gene deletions have been reported across malaria endemic countries in Africa with the highest prevalence of

deletion from Eritrea (62%) [14] and the lowest from Angola (0.4%) [19]. Indeed, in some hospitals in Eritrea the levels of gene deletions were as high as 80% [14]. As the malaria transmission intensity and intervention history in Ethiopia is similar to that in Eritrea, it is therefore possible that *pfhrp*2/3 deleted isolates may be in circulation in Ethiopia at similarly high frequencies. Following the WHO recommendation for *pfhrp*2/3 surveys and cross border surveillance activities, this study investigates the extent of *pfhrp*2/3 deleted *P. falciparum* parasites in an Ethiopian *P. falciparum* population. Molecular analyse targeting the region across exons and flanking genes were used to provide evidence of gene deletions in the *pfhrp*2/3 genes.

## Materials and methods

### Study area

The study was conducted in Adama town, East Shoa Zone, Oromia, Ethiopia. The town is located at 8.54˚N and 39.27˚E, at an elevation of 1,712 meters above sea levels and is 99 km southeast of Ethiopia's capital, Addis Ababa. Located between the base of an escarpment in the West and the Great Rift Valley in the East, Adama town experiences rainfall from mid-June to mid-September with short rains in March. Adama Malaria diagnostic center is the oldest laboratory exclusively committed to malaria diagnosis. As a matter of fact, people from the Adama town and the surrounding rural areas preferentially use this laboratory as far as malaria diagnosis is concerned over hospitals and other surrounding health centers in the town. The study site exhibits high malaria transmission with both *P.falciparum* and *P.vivax* malaria are co-endemic. In the study area, major and minor transmission seasons exist. The major malaria transmission season is from September through November and the minor from April to May. *Anopheles arabiensis* is the dominant malaria vector.

**Sample collection and diagnosis of malaria.**   The study was initiated to explore the genetic variation and deletions in the *pfhrp2/3* genes. Finger-prick blood samples were collected from 64 febrile patients attending Adama Malaria Diagnostic Centre from September through December 2015. Thick and thin blood smears were prepared for microscopic diagnosis of malaria parasite infections and identification of species. Parasite densities were calculated according to described standard methods (Parasites/μL = no. of asexual parasites X 8000/no. of WBC counted) [1]. Infected blood samples were spotted onto Whatman 3MM filter papers for parasite DNA extraction.

### PCR confirmation of *Plasmodium falciparum* infections

Parasite DNA was extracted from dried blood spots using the chelex100 extraction method as described earlier [20]. The presence of *Plasmodium* species was confirmed by targeting 18S rRNA by a nested polymerase chain reaction (nPCR) using genus-specific primers rPLU 6: (5′ TTAAAATTGTTGCAGTTAAAACG3′), rPLU 5: (5′ CCTGTTGTTGCCTTAAACTTC3′) followed by species-specific primers rFAL 1: (5′ TTAAACTGGTTTGGGAAAACCAAATAT ATT3′), rFAL2: (5′ ACACAATGAACTCAATCATGACTACCCGTC3′) as described by Snounou [21]. The cycling conditions were as follows: denaturation, 95˚C for 5 min; 35 cycles of 94˚C for 30 s, 56˚C for 30 s, and 60˚C for 60 s; and a final extension at 60˚C for 5 min. The presence of amplification product is detected by simple ethidium bromide staining following agarose gel electrophoresis and a 205 bp size of the PCR product confirms *P. falciparum*.

**PCR-based genotyping of *pfhrp*2/3 deletions.**   Amplifications of exons 2 and their flanking regions of *pfhrp*2/3 genes were done by semi-nested PCR [22] using published protocols and primers (Table 1). Nest-1 PCR targets repeat sequences within the most variable part of the genes while the second primer set targets an intron region. For *pfhrp*2, PCR nest-1 product sizes of 720–830 bp were expected while for *Pfhrp*3, the expected PCR product size was< 500

**Table 1. Primers name and sequences used to amplify *pfhrp*2/3 genes including the flanking regions.**

| Primer name | Gene | Sequence (5' → 3') |
|---|---|---|
| Hrp-2 outer (reverse primer) | HRP2 | 5'-TCT ACA TGT GCT TGA GTT TCG-3' |
| Hrp-2 outer (forward primer) | HRP2 | 5'¯GGT TTC CTT CTC AAA AAA TAA AG-3' |
| Hrp-2 inner (forward primer) | HRP2 | 5'¯ GTA TTA TCC GCT GCC GT TTT GCC-3' |
| Hrp-2 inner (reverse primer) | HRP2 | 5'¯CTA CAC AAG TTA TTATTA AAT GCG GAA- 3' |
| *pf*hrp2newoutfw | HRP2 | Fw: ATA TTT GCA CAT CTT GC |
| *pf*hrp2newoutrev | HRP2 | Rev: ATG GTT TCC TTC TCA AA |
| *pf*hrp2newnestfw | HRP2 | Fw: TCG CTA TCC CAT AAA TTA CA |
| *pf*hrp2newnestrev | HRP2 | Rev: GAT TAT TAC ACG AAA CTC AAG C |
| 228 outer-forward | 228 | Fw: CAA TAG TTG CTT GTG CGG ATG |
| 228 outer-reverse | 228 | Rev AGA AGT TGC AGA GAC ATA CTT AGG |
| 228 nested-forward | 228 | Fw: AGA CAA GCT ACC AAA GAT GCA GGT |
| 228 nested-reverse | 228 | Rev: TAA ATG TGT ATC TCC TGA GGT AGC |
| 230 outer-forward | 230 | Fw: CCC TGC TAT ATA GAT GAG GAA A |
| 230 outer-reverse | 230 | Rev: CTA CCA CTT CTG TTG CTA CC |
| 230 nested forward | 230 | Fw: TAT GAA CGA AAT TTA AGT GAG GCA |
| 230 nested-reverse | 230 | Rev: TAT CCA ATC CTT CCT TTG CAA CAC C |
| Hrp3 out rev new | HRP3 | 5´-CCA TAC ACT TAT GCT GTA TTTA- 3´ |
| Hrp3 outfw new | HRP3 | 5´- TGG TAA TTT CTG TGT TTA TG- 3´ |
| Hrp3-2 nestfw | HRP3 | 5´- TAT CCG CTG CCG TTT TTG CTT CC- 3´ |
| Hrp3 nest rev | HRP3 | 5´- TGG TGT AAG TGA TGC GTA GT- 3´ |
| MAL 475 REV set1 (out-rev) | 475 | 5´-TCC CAC ATC GTA TAT CTC AGT TTC- 3´ |
| MAL 475 FWD set1 (out-fw) | 475 | 5´-GGA AAG CAC AAC AAG ATG GAT AC- 3´ |
| MAL 13PI 475 rev (nest-rev) | 475 | 5´-TCG TAC AAT TCA TCA TAC TCA CC- 3´ |
| MAL 13PI 475 fw (nest-fw) | 475 | 5´-TTC ATG AGT AGA TGT CCT AGG AG- 3´ |
| MAL 485 REV set1 (out-rev) | 485 | 5´-GCT TCT TTC CAC ATT TCT CAC AT- 3´ |
| MAL 485 FWD set5 (out-fw) | 485 | 5´-GTG TGT TTC CAT GTA TTA CGG AAG- 3´ |
| MAL 12PI 485 rev (nest-rev) | 485 | 5´-AAA TCA TTT CCT TTT ACA CTA GTG C- 3´ |
| MAL 12PI 485 fw (nest-fw) | 485 | 5´-TTG AGT GCA ATG ATGATG GGA G- 3´ |

bp. The PCR products were purified by the GeneJet PCR Cleanup Kit from Thermo Fisher Scientific and sent for sequence determination at Eurofins genomics, Germany. Sequences were analysed by the 4peaks program (A. Griekspoor and Tom Groothuis, nucleobytes.com).

## Amplification of *pfhrp2/3* flanking regions

For amplifications of genes immediately flanking *pfhrp*2 (MAL7P1.230 (5.535 kb upstream) and MAL7P1.228 (6.49 kb downstream)), and *pfhrp*3 (MAL13P1.485 (4.404 kb upstream) and MAL13P1.475 (1.684 kb downstream)), the following primers and PCR conditions were used (Table 2).

To rule out the possibility that the absence of amplification in *pfhrp*2/3 may be an artifact of the PCR, alternative primers with different binding sites and amplification conditions were used. All primers were used on samples from Tanzania with intact *pfhrp*2/3 genes with good results. In addition, amplifications of *pfmdr*1 and *pfubp*-1 genes were successful in all samples suggesting the deletion of *pfhrp*2/3 genes.

**Ethical issue.** The study was approved by Aklilu Lemma Institute of Pathobiology, Addis Ababa University, Institutional review Board. Written consent and/or assent were obtained from each study participant.

**Table 2. PCR conditions and expected product sizes of the *pfhrp*2/3 flanking regions.**

| Gene | PCR conditions | Expected PCR product size |
|---|---|---|
| MAL7P1_228 | 94˚C for 10 min, followed by 94˚C for 30 sec, 60˚C for 30 sec, 68˚C for 1 min | 227 bp |
| MAL7P1_230 | 94˚C for 10 min, followed by 94˚C for 30 sec, 60˚C for 30 sec, 68˚C for 1 min | 346 bp |
| MAL13P1_475 | 94˚C for 10 min, followed by94˚Cfor 30 sec, 60˚C for 30 sec, 68˚C for 1 min | 260 bp |
| MAL12P1_485 | 94˚C for 10 min, followed by 94˚C for 30 sec, 60˚C for 30 sec, 68˚C for 1 min | 287 bp |

## Results

### PCR confirmation of *Plasmodium falciparum* infections

Of 189 self-reporting febrile patients seeking malaria diagnosis at Adama Malaria Diagnostic Centre, 33.9% (64/189) were positive for *P. falciparum* as confirmed by expert microscopy which was later proven positive by PCR. The male: female ratio was 3.1:1. Participant's mean age was 25.2 years (range 11–48). The minimum parasite density reported was 400 parasites/ μL. Although all microscopically confirmed cases tested positive by PCR, only 50 samples had enough DNA for further analysis of deletion in *pfhrp*2/3 genes and the flanking regions.

By targeting six regions in the *pfhrp*2/3 genes and their flanking genes, different deletion patterns were observed in Ethiopian *P. falciparum* clinical samples. Most parasite isolates had deleted the gene located 3' of *pfhrp*2, PF3D7_0831900, compared to the flanking gene5', PF3D7_0831700. In contrast, the 5' flanking PF3D7_1372100 gene, upstream of *pfhrp*3, showed more deletions than the downstream 3' flanking PF3D7_1372400 region. Combining deletions in the genes and flanking regions, the most common pattern exhibited in the isolates was the presence of the two flanking regions for *pfhrp*2 in combination with the downstream flanking region for *pfhrp*3. This was followed by isolates that had a deleted downstream flanking region of *pfhrp*3 but with the two flanking regions of *pfhrp*2 retained. Notably, only one isolate showed intact flanking regions for both gene loci. Amplifications of *pfmdr*1 and *pfubp*-1 genes in these samples are an indication that the absence of PCR products in the *pfhrp*2/3 genes and the respective flanking regions are due to deletions.

**Genetic deletion of *pfhrp*2 and *pfhrp*3 and their flanking genes.** The deletion variant at *pfhrp2* gene was fixed in the population analysed as the gene was deleted in all 50(100%) isolates assessed. The deletion extended downstream *pfhrp*2 gene flanking region towards the Pf3D7 0831900(MAL7PI.230) gene in 11/50 (22%) of the cases (Table 3). In contrast, only 2/50 (4%) of samples had deletions for the upstream gene Pf3D7 0831700(MALPI.228).

Similar results were observed for *pfhrp*3 and flanking regions. Here, all the isolates had deletions in the *pfhrp*3 gene (100%). Like for *pfhrp*2 gene, *pfhrp*3 deletion extended to the downstream flanking region to include Pf3D7 13272400 (MAL13PI.485) in 40% of samples. However, the extension of the deletion was more prevalent upstream towards Pf3D7 081372100 (MAL13PI.475), with 49/50 (95%) of isolates deleted at these loci.

The summary of deletions in *pfhrp*2 and *pfhrp*3 genes and the respective flanking regions are indicated in S1 File.

In addition to the exon primers that cover the normally analysed variable region, a set of primers targeting an intron sequence with a varying number of AT repeats. In spite of the negative results for all samples in exon-based PCR, five samples actually gave PCR products for the intron region. The samples contained different numbers of AT repeat sequence motif (one

**Table 3. Extension of deletions of *pfhrp* 2 and *pfhrp*3 genes, the respective flanking regions and exon primers used in Ethiopian isolates.**

| DNA sample ID. | Gene 228 | *pfhrp2* | Gene 230 | Gene 475 | *pfhrp3* | Gene 485 |
|---|---|---|---|---|---|---|
| 1 | + | - | - | - | - | + |
| 2 | + | - | - | - | - | + |
| 3 | + | - | + | - | - | - |
| 4 | + | - | + | - | - | + |
| 5 | + | - | + | - | - | + |
| 6 | + | - | + | - | - | + |
| 7 | + | - | + | - | - | + |
| 8 | + | - | + | (+) | - | + |
| 9 | + | - | + | - | - | + |
| 10 | + | - | + | - | - | + |
| 11 | + | - | + | - | - | + |
| 12 | + | - | + | - | - | - |
| 13 | + | - | + | - | - | - |
| 14 | + | - | - | - | - | - |
| 15 | + | - | + | - | - | + |
| 16 | + | - | - | - | - | + |
| 17 | + | - | - | - | - | + |
| 18 | + | - | - | - | - | + |
| 19 | + | - | + | - | - | + |
| 20 | + | - | + | - | - | - |
| 21 | + | - | + | - | - | - |
| 22 | + | - | + | - | - | - |
| 23 | + | - | + | - | - | - |
| 24 | + | - | + | - | - | - |
| 25 | + | - | - | - | - | - |
| 26 | + | - | + | - | - | + |
| 27 | + | - | + | - | - | + |
| 28 | + | - | - | - | - | + |
| 29 | + | - | + | - | - | + |
| 30 | + | - | + | - | - | - |
| 31 | + | - | + | - | - | - |
| 32 | + | - | + | - | - | - |
| 33 | - | - | + | - | - | - |
| 34 | - | - | + | - | - | - |
| 35 | + | - | + | - | - | - |
| 36 | + | - | - | - | - | - |
| 37 | + | - | + | - | - | + |
| 38 | + | - | + | - | - | - |
| 39 | + | - | - | - | - | + |
| 40 | + | - | - | - | - | + |
| 41 | + | - | + | - | - | + |
| 42 | + | - | + | - | - | + |
| 43 | + | - | + | - | - | + |
| 44 | + | - | + | - | - | + |
| 45 | + | - | + | - | - | + |
| 46 | + | - | + | - | - | - |
| 47 | + | - | + | - | - | - |

(*Continued*)

**Table 3.** (Continued)

| DNA sample ID. | Gene 228 | *pfhrp2* | Gene 230 | Gene 475 | *pfhrp3* | Gene 485 |
|---|---|---|---|---|---|---|
| 48 | + | - | + | - | - | - |
| 49 | + | - | + | - | - | + |
| 50 | + | - | + | - | - | + |

sample with 10 repeats, and four samples with 17 repeats) and suggest that the deletion in this region did not involve the entire region (Table 4).

## Discussion

As per WHO which hosted a technical consultation on *pfhrp*2/3 gene deletions and drafted interim guidance for investigating false-negative RDTs [23], understanding the distribution and evolution of these mutant parasites is a priority. However, it is yet unknown whether reliance on *Pf*HRP2-based RDTs to guide treatment across malaria endemic countries is exerting evolutionary pressure favouring the spread of this mutation. At present, PfHRP2-based RDTs are central to malaria control programmes in spite of the threat by parasites that do not express *Pf*HRP2. This study is the first to report the presence of extensive deletion of *pfhrp*2 gene including deletion in its structural homolog, *pfhrp*3, in clinical isolates in Ethiopia

All 50 samples (100%) yielded deletion for *pfhrp*2 and *pfhrp*3 genes. In *pfhrp*2 gene, the deletion extended to 4% (2/50) of flanking region gene 228. Flanking region gene 230 contained deletions in 22% (11/50) of the samples. Around *pfhrp*3 genes 475 and 485 were deleted for 95% and 40% of the samples, respectively. The fact that HRP2-based RDTs tests accounted for 74% of malaria diagnostic testing in the sub-Saharan Africa in 2017 [24], such massive utilization of RDTs could lead to selection and spread of *P. falciparum* strains that can evade detection through the deletion of the *pfhrp*2 genes.

Nowadays, a great concern with the use of *Pf*HRP2-based RDTs malaria diagnosis has been the evolving reports of *P. falciparum* isolates lacking the *pfhrp*2 and *pfhrp*3 genes, which respectively encode the *Pf*HRP2 and the PfHRP3 proteins [12,25,26]. The deletion assay includes six targets of *pfhrp*2/3 and flanking regions, and at least one locus was amplified for 50 samples out of 64 microscopically and PCR confirmed *P. falciparum* clinical samples collected. While the deletion in *pfhrp*2 gene extended downstream in 11/50 (22%) of the isolates, the deletion was only 4% in upstream of *pfhrp*2. However, it is unclear if these *pfhrp*2 deletions are recent events or emerged prior to the introduction of *Pf*HRP2-based RDTs in Ethiopia. In Peru, for instance, *pfhrp2*-deleted parasites were present before the introduction of RDTs, but the sweep in the population that occurred after RDT introduction shows the strength of

**Table 4.** Repeat sequences in *pfhrp*2 intron, in 50 Ethiopian samples tested, those not listed did not give PCR products.

| Sample ID. (No. tested = 50 samples) | AT repeats intron |
|---|---|
| 2 | 17 |
| 24 | 17 |
| 25 | 10 |
| 27 | 17 |
| 34 | 17 |

selection against this new diagnostic tool [27]. It is indicated that the extensive use of *Pf*HRP2-based RDTs is sufficient to select *P. falciparum* parasites lacking this protein [15].

Similarly, deletion was evident in 100% of the isolates analysed in *pfhrp*3 gene and extended to the downstream flanking region that codes for Pf3D7 13272400 (MAL13PI.485) gene in 40% of the isolates. The *pfhrp*3 deletion also extended upstream to Pf3D7 081372100 (MAL13PI.475) region in 49/50 (95%) of the isolates, exhibiting complete absence of the locus. Two distinct repeat motifs were observed for *pfhrp*2 intron region in 50 of the samples tested suggesting that the deletion in this region did not involve the entire region.

Interestingly, *Pf*HRP2-based RDTs are far more popular in Ethiopia than pLDH-based RDTs, partly because of their higher sensitivity for *P. falciparum* diagnosis. The presence of deletions in both *pfhrp*2 and *pfhrp*3 genes suggest that this could have been a result of a recent selection as a consequence of the widely used *Pf*HRP2-based RDTs. The possible spread of *pfhrp*2/3 deleted parasite from a neighbour country like Eritrea can't be overlooked for the presence high deletions in Ethiopian *P. falciparum* populations. Very high frequencies of these deletions have also been reported in Eritrea, a close geographic population and neighbour to Ethiopian parasite populations [13,14]. Deletions in both genes are less frequent in other African populations, though this phenomenon is quite prevalent in South American countries [27,28]. Recent whole genome analysis of *P. falciparum* across Africa countries found isolates from Ethiopia to be highly divergent from the rest of continent, defining a genomic background that could respond differently to selective forces such as RDTs and drugs [20]. These deletion isolates formed a closely related cluster probably from clonal proliferation of a recent *pfhrp*2-deleted ancestor [27]. Expansion of these deleted isolates could jeopardise the effectiveness of *Pf*HRP2-based RDTs.

RDT-based malaria diagnosis followed by treatment could be selectively clearing infections with parasites retaining the *pfhrp*2/3 genes and hence increase the rate of spread of parasites with deletions [29]. When the *pfhrp*2 gene deletion was reported in 2010 in South America, this led to the recommendation against the use of *Pf*HRP2-based RDTs in these areas [30–32]. If the results here are corroborated in a larger study across Ethiopia, a similar recommendation may be warranted. Unfortunately, patient recruitment in this study was based on microscopy and this does not allow us to determine the outcome of *Pf*HRP2/3-based RDT for *P. falciparum* population with complete deletion of *pfhrp*2/3 genes. Hence, further large-scale studies using microscopy and *Pf*HRP2/3-based RDTs are required to validate these high frequencies of *pfhrp*2/3 gene deletions and their effect on RDT malaria diagnosis in Ethiopia. For now, polymorphisms in *pfhrp*2/3 genes in Ethiopian isolates don't seem to influence performance of currently used *Pf*HRP2 RDTs given that they have been widely used in the country. Furthermore, as the samples were collected from one location in Ethiopia, a geographically expanded study would better inform the national malaria control program (NMCP) on need for reviewing policy on type of mRDTs in the country and the extent of *pfhrp*2/3 genes deletions.

This study has several important limitations. The samples were collected during a single malaria transmission season spanning September through December 2015 from one study site. Hence, the results here can't be generalized to the clinical isolates from other endemic areas of Ethiopia. The number of isolates analysed was also small in number (50 *P. falciparum* clinical isolates). The fact that the clinical samples were collected using microscopy alone, it is impossible to know if the deleted isolates would test negative or positive for RDTs in the absence of *pfhrp*2/3 genes as we didn't perform RDT-based diagnosis.

In summary, *P. falciparum* parasite populations with deletions of the *pfhrp*2 and *pfhrp*3 genes are present in Adama site of Ethiopia. Continuous monitoring of deletions among clinical isolates in the target regions is important in this era of malaria elimination, which largely depends on RDTs.

## Supporting information

**S1 File. Summary of *pfhrp*2, *pfhrp*3 amplification and their respective flanking genes in *P. falciparum* samples collected in Ethiopia.**
(PDF)

**S2 File. PCR protocol used for amplification of *Plasmodium* DNA.**
(DOC)

**S3 File. *Pfhrp2/3* gene amplification for Tanzanian samples (both positive and negative) and Ethiopian samples (all negatives).** a). Tanzanian samples tested positives for hrp2/3 genes. b). Ethiopian samples tested negatives for hrp2/3 genes (please note that no positive control is included).
(DOC)

## Acknowledgments

We sincerely thank all study for their participations and laboratory technicians for their support and cooperation during this study.

## Author Contributions

**Conceptualization:** Lemu Golassa, Gote Swedberg.

**Formal analysis:** Alebachew Messele, Alfred Amambua-Ngwa.

**Methodology:** Gote Swedberg.

**Writing – original draft:** Lemu Golassa, Alebachew Messele, Gote Swedberg.

**Writing – review & editing:** Alfred Amambua-Ngwa.

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
