## [Decision Letter · Decision Letter 0]

4 Sep 2020

PONE-D-20-24429

High prevalence and extended deletions in Plasmodium falciparum hrp2/3 genomic loci in Ethiopia

PLOS ONE

Dear Dr. Golassa,

Thank you for submitting your manuscript to PLOS ONE. After careful consideration, we feel that it has merit but does not fully meet PLOS ONE’s publication criteria as it currently stands. Therefore, we invite you to submit a revised version of the manuscript that addresses the points raised during the review process.

I am sure that the reviewers' comments are very useful to improve this manuscript. Please consider and respond to all of the expert reviewers' comments and extensively revise the manuscript based on their comments. 

We look forward to receiving your revised manuscript.

Kind regards,

Takafumi Tsuboi

Academic Editor

PLOS ONE

Journal Requirements:

2. Thank you for including the following funding information within the acknowledgements section of your manuscript; "This work was supported through the DELTAS Africa Initiative [DELGEME grant 107740/Z/15/Z]. The DELTAS Africa Initiative is an independent funding scheme of the African Academy of Sciences (AAS)’s Alliance for Accelerating Excellence in Science in Africa (AESA) and supported by the New Partnership for Africa’s Development Planning and Coordinating Agency (NEPAD Agency) with funding from the Wellcome Trust [DELGEME grant 107740/Z/15/Z] and the UK government. "

"No, the funders had no role in study design, data collection and analysis, decision to publish, or preparation of the manuscript."

3. Thank you for including your competing interests statement;

"No, I have read the journal's policy and the authors of this manuscript have the following competing interests"

Reviewers' comments:

Reviewer's Responses to Questions

**Comments to the Author**

1. Is the manuscript technically sound, and do the data support the conclusions?

Reviewer #1: Partly

Reviewer #2: Yes

2. Has the statistical analysis been performed appropriately and rigorously? 

Reviewer #1: N/A

Reviewer #2: Yes

3. Have the authors made all data underlying the findings in their manuscript fully available?

Reviewer #1: Yes

Reviewer #2: Yes

4. Is the manuscript presented in an intelligible fashion and written in standard English?

Reviewer #1: Yes

Reviewer #2: Yes

5. Review Comments to the Author

Reviewer #1: In this study, the authors determined the deletions of pfhrp2 and pfhrp3, as well as of genes upstream and downstream of these two genes, by PCR analysis of samples from patients infected with Plasmodium falciparum in Ethiopia. The authors show that in a substantial number of samples, the parasite lacks both pfhrp2 and pfhrp3, which may lead to pseudo-negative results when using HRP2-based RDTs for testing individuals suspected to have malaria in Ethiopia. I think some modifications and additional explanations are required.

(1) Abstract (Conclusion, line 5 from the bottom): Because the term “fixed” might have the same meaning as “repaired,” I was confused when I read the manuscript for the first time. In this paper, did you mean “found?” I think it would be better to rephrase the sentence.

(2) Introduction (Paragraph 3, line 9): Please replace “pfhr2/3loci” with “pfhrp2/3 loci.”

(3) Introduction (Paragraph 4, lines 1 to 6 from the bottom): The sentences from “As per” to “of this mutation” should be moved to the Discussion section.

(4) Methods: I think the information about the pseudo-positive rates of the HRP2-based RDTs is important. Do you have results of the RDTs for samples from the 50 patients infected with hrp2- hrp3- parasite?

(5) Results (p. 14, line 12): The term “fixed” should be replaced with an appropriate term.

(6) Results (p. 14, line 4): You have annotated “Table 3;” however, important results and/or information are included in Additional file 1. You should annotate Additional file 1 as Table 3.

(7) Table 3: I cannot understand what have you listed in Table 3. Why did you list the five samples?

Reviewer #2: This manuscript reports the results of a small study assaying the prevelence of Pfhrp2 and 3 deletions in 50 samples from a single collection point in Ethiopia collected between September and November 2015. The authors report that all parasites characterised had deletions of both genes, presumably rendering them non-detectable by the commonly used PfHRP2 based RDTs used in this area. The methodology is appropriate, and the results of interest. I have several points that the authors may consider for revision:

As there are no page numbers on the document, I have numbered them myself, with Page 1 of the PDF being the abstract…

General

The manuscript needs a thorough proofreading to correct numerous typological, spelling and grammatical errors.

Please could data on the malaria endemicity of the region from which samples were collected be given? Is this a high transmission region for P. falciparum?

Samples are rather few, and are from a very specific region of Ethiopia. It is very difficult to ascertain whether the results here are representative of a the wider Ethiopian P. falciparum population or simply constitute a small local population. There is no description of the diagnosistics commonly used in this region of Ethiopia – as the patients here were diagnosed by microscopy, is it right to assume this is the standard protocol? In which case, presumably, RDTs are not used here. In which case, the authors need to offer some hypothesis of why HRP2 deleted parasites dominate in a region where there should have not undergone selection….

There is a general lack of essential information in the introduction; it would be beneficial for the reader to know not only the malariometrics of the study area (entomological inoculation rates, malaria parasite species present, parasite prevelence etc), but also the extent to which RDTs are used for diagnosis, and when they were introduced.

Introduction

Page 2, sentence 1: please use percentages (60%) instead of “in 0.60 of cases”.

Page 2, sentence 3. 40% reduction since when?

Page 2, paragraph 3. The authors contend that PfHRP2 deletions have not been extensively surveyed in Africa. There are, in fact, very many reports detailing such surveys.

Methods

Page 4. Please at least give the primer names and conditions for the PCR rather than referring to previous publications

Page 6. The positive control samples (with the Pfhrp2 genes intact) come from Tanzania. Is it possible that the strains circulating in Ethiopia have different sequences at the primer binding sights in hrp2/3 genes, rendering the PCR ineffective? Why wasn’t an Ethiopian PfHRP2 intact P. falciparum used as a positive control?

Discussion

Page 11 paragraph 1 – the authors find three deletion variants according to the size of the deletion. This does not fit with proposal of clonal expansion of one mutant.

6. PLOS authors have the option to publish the peer review history of their article (what does this mean?). If published, this will include your full peer review and any attached files.

Reviewer #1: No

Reviewer #2: **Yes: **Richard Culleton

---

## [Author Response · Author response to Decision Letter 0]

15 Oct 2020

Dear respected reviewers,

I would like to thank you for your critical review of our manuscript and constructive comments you have given us to enrich our manuscript.

Dear Editor,

I appreciate your constructive comments and guidance.

I have uploaded a point-by-point response to your respective comments online for your consideration

Sincerely,

Lemu Golassa, PhD

---

## [Decision Letter · Decision Letter 1]

19 Oct 2020

PONE-D-20-24429R1

High prevalence and extended deletions in Plasmodium falciparum hrp2/3 genomic loci in Ethiopia

PLOS ONE

Dear Dr. Golassa,

Thank you for submitting your manuscript to PLOS ONE. After careful consideration, we feel that it has merit but does not fully meet PLOS ONE’s publication criteria as it currently stands. Therefore, we invite you to submit a revised version of the manuscript that addresses the points raised during the review process.

Both expert reviewers have mostly appreciated the Authors' efforts for the significant improvement of the manuscript. However, please make a minor revision with all the recommendation by the Reviewer 2. Especially, “fixed” is entirely appropriate so please revert "found" back to "fixed".

We look forward to receiving your revised manuscript.

Kind regards,

Takafumi Tsuboi

Academic Editor

PLOS ONE

Reviewers' comments:

Reviewer's Responses to Questions

**Comments to the Author**

1. If the authors have adequately addressed your comments raised in a previous round of review and you feel that this manuscript is now acceptable for publication, you may indicate that here to bypass the “Comments to the Author” section, enter your conflict of interest statement in the “Confidential to Editor” section, and submit your "Accept" recommendation.

Reviewer #1: All comments have been addressed

Reviewer #2: (No Response)

2. Is the manuscript technically sound, and do the data support the conclusions?

Reviewer #1: Yes

Reviewer #2: Yes

3. Has the statistical analysis been performed appropriately and rigorously? 

Reviewer #1: N/A

Reviewer #2: I Don't Know

4. Have the authors made all data underlying the findings in their manuscript fully available?

Reviewer #1: Yes

Reviewer #2: Yes

5. Is the manuscript presented in an intelligible fashion and written in standard English?

Reviewer #1: No

Reviewer #2: Yes

6. Review Comments to the Author

Reviewer #1: The sentence on p. 10, line 3 (In addition to ~~ AT repeats) was incomplete. “a set of primers targeting an intron sequence with a varying number of AT repeats was investigated.”?

Reviewer #2: Abstract

The word “fixed” is generally understood to mean that an allele is at 100% prevalence in a population; it is a well-known and understood genetic term. In my opinion, the authors should revert to the word “fixed”, rather than using the replacement term “found” which doesn’t really make sense in the context of this sentence.

In any incidence where the prevalence of a mutation was found to be 100%, the word “fixed” is entirely appropriate, and preferred over the term “found”.

General

In response to my comment regarding the usage of RDTs as diagnostics in the area, the authors have answered, convincingly, in their replies to the reviewers document, but have not included this information in the modified manuscript! It would be useful for readers of the MS to also have this information. Please could they include this information in the discussion section?

7. PLOS authors have the option to publish the peer review history of their article (what does this mean?). If published, this will include your full peer review and any attached files.

Reviewer #1: No

Reviewer #2: **Yes: **Richard Culleton

---

## [Author Response · Author response to Decision Letter 1]

19 Oct 2020

Dear Editor,

Thanks you very much for the update.

Query 1. Both expert reviewers have mostly appreciated the Authors' efforts for the significant improvement of the manuscript. However, please make a minor revision with all the recommendation by the Reviewer 2. Especially, “fixed” is entirely appropriate so please revert "found" back to "fixed".

Response: Thanks, we revert ‘fixed’ instead of ‘found’ across the document. Other points raised by reviewer # have been well addressed in the previous responses to reviewers.

---

## [Editor Report · Decision Letter 2]

21 Oct 2020

High prevalence and extended deletions in Plasmodium falciparum hrp2/3 genomic loci in Ethiopia

PONE-D-20-24429R2

Dear Dr. Golassa,

We’re pleased to inform you that your manuscript has been judged scientifically suitable for publication and will be formally accepted for publication once it meets all outstanding technical requirements.

Kind regards,

Takafumi Tsuboi

Academic Editor

PLOS ONE
---

## [Editor Report · Acceptance letter]

26 Oct 2020

PONE-D-20-24429R2 

High prevalence and extended deletions in *Plasmodium falciparum hrp2/3* genomic loci in Ethiopia 

Dear Dr. Golassa:

I'm pleased to inform you that your manuscript has been deemed suitable for publication in PLOS ONE. Congratulations! Your manuscript is now with our production department. 

Kind regards, 

on behalf of

Prof. Takafumi Tsuboi 

Academic Editor

PLOS ONE